# Biomimetic Proteoglycans for Intervertebral Disc (IVD) Regeneration

**DOI:** 10.3390/biomimetics9120722

**Published:** 2024-11-22

**Authors:** Neha Chopra, James Melrose, Zi Gu, Ashish D. Diwan

**Affiliations:** 1Spine Service & Spine Labs, St George & Sutherland School of Clinical Medicine, Faculty of Health and Medicine, University of New South Wales, Kogarah, NSW 2217, Australia; n.chopra@unsw.edu.au; 2Graduate School of Biomedical Engineering, University of New South Wales, Sydney, NSW 2052, Australia; james.melrose@sydney.edu.au; 3Raymond Purves Laboratory, Institute of Bone and Joint Research, Kolling Institute of Medical Research, Northern Sydney Local Health District, Royal North Shore Hospital, St. Leonards, NSW 2065, Australia; 4Sydney Medical School, University of Sydney at Royal North Shore Hospital, St. Leonards, NSW 2065, Australia; 5NanoBiotechnology Research Group, School of Chemical Engineering, Faculty of Engineering, University of New South Wales, Sydney, NSW 2052, Australia; zi.gu1@unsw.edu.au; 6Australian Centre for NanoMedicine, University of New South Wales, Sydney, NSW 2052, Australia; 7UNSW RNA Institute, University of New South Wales, Sydney, NSW 2052, Australia; 8Discipline of Orthopaedic Surgery, Royal Adelaide Hospital and University of Adelaide, Adelaide, ADL 5005, Australia

**Keywords:** biomimetic proteoglycans, intervertebral disc, aggrecan

## Abstract

Intervertebral disc degeneration, which leads to low back pain, is the most prevalent musculoskeletal condition worldwide, significantly impairing quality of life and imposing substantial socioeconomic burdens on affected individuals. A major impediment to the development of any prospective cell-driven recovery of functional properties in degenerate IVDs is the diminishing IVD cell numbers and viability with ageing which cannot sustain such a recovery process. However, if IVD proteoglycan levels, a major functional component, can be replenished through an orthobiological process which does not rely on cellular or nutritional input, then this may be an effective strategy for the re-attainment of IVD mechanical properties. Furthermore, biomimetic proteoglycans (PGs) represent an established polymer that strengthens osteoarthritis cartilage and improves its biomechanical properties, actively promoting biological repair processes. Biomimetic PGs have superior water imbibing properties compared to native aggrecan and are more resistant to proteolytic degradation, increasing their biological half-life in cartilaginous tissues. Methods have also now been developed to chemically edit the structure of biomimetic proteoglycans, allowing for the incorporation of bioactive peptide modules and equipping biomimetic proteoglycans as delivery vehicles for drugs and growth factors, further improving their biotherapeutic credentials. This article aims to provide a comprehensive overview of prospective orthobiological strategies that leverage engineered proteoglycans, paving the way for novel therapeutic interventions in IVD degeneration and ultimately enhancing patient outcomes.

## 1. Introduction

### 1.1. Background of Low Back Pain

Epidemiology: A 10-year global study of all major musculoskeletal disorders has shown that LBP is the most consequential musculoskeletal condition in terms of the resultant years lived with disability and the socioeconomic impact it produces [1,2,3,4,5]. Intervertebral disc degeneration (IVDD) is considered to affect ~80% of the general population, leading to LBP of sufficient severity to warrant consultation with a physician [6] and resulting in a loss of productive work days [3]. The incidence tends to increase with age, with higher rates observed in individuals aged 30–50 years, but it can affect all age groups. It is estimated conservatively that 619 million people are affected by LBP globally [4,5,7]; an increased incidence of LBP in the fifth and sixth decades [8] and the advancing age of the global general population [9] show that this burden will only increase in severity over the next two decades. Comprehensive guidelines and bulletins published by the World Health Organization (WHO) [6], the International Association for the Study of Pain (IASP) [10], the National Institutes of Health (NIH) [11], the World Bank [12], the Australian Institute of Health and Welfare (AIHW) [13], and the United Nations (UN) [14] confirm this conclusion.

Risk Factors: Risk factors for LBP can be categorized into several groups. Demographically, LBP tends to increase, with age with women reporting higher prevalence rates [15]. Occupational factors like heavy lifting jobs, repetitive motions, and prolonged sitting may contribute to the early onset of LBP. This coupled with lifestyle factors such as obesity and smoking can progressively lead to degenerative disc diseases (DDDs). Research also suggests that psychosocial factors such as stress, anxiety, and depression can increase the risk of developing LBP [16]. Genetically, a family history of predisposition to DDDs, a history of a previous injury and anatomical variations can all lead to an increased risk of pain [17].

Public health implications: LBP has been costed in the UK at GBP 12.3 billion [18] and AUD 9.17 billion in Australia [19]. The American Academy of Pain Medicine published annual costs in 2006 for chronic pain of USD 560 to 635 billion, noting that 53% of all chronic pain patients in the USA were affected by LBP [20]. The direct medical and indirect costs of LBP are more than USD 50 billion per annum and could be as high as USD 100 billion [21,22]. In 2006, a review of total costs associated with LBP in the United States indicated that they exceeded USD 100 billion per year [23]. Annual costs for the treatment of LBP to the UK National Health Service of GBP 5 billion have been published, while in the USA, they were costed at USD 134 billion in 2016 [24].

Patho mechanics of LBP: LBP is a musculoskeletal disorder [2,25,26,27] elicited by stimuli originating in the spinal muscles, vertebral body, anterior and posterior longitudinal ligaments, facet joint cartilage and its capsular synovial tissues, and the IVD; however, the IVD is considered a major contributor to the generation of LBP [20,26,28]. LBP has been categorized as mechanical, non-mechanical, or due to referred pain [29,30]. Neuropathic pain is caused by inflammation and irritation of neural tissue and can be distinguished from nociceptive pain, which is the body’s reaction to painful stimuli from back muscle or nerve damage per se [29]. Trauma to the PNS/CNS can lead to pain hypersensitivity, with neuropathic pain persisting in some tissues for prolonged periods.

Intervertebral Disc Degeneration: Many studies show that intervertebral disc degeneration (IVDD) is a major contributor to low back pain (LBP) [25,26,27] due to its inability to withstand normal axial spinal loading. Multiple anatomical structures in the spine besides the IVD can contribute to pain responses including the cartilaginous endplate, paradiscal myotendinous tissues, and osteoarthritic facet joint articular cartilage [31]. Degenerative changes in spinal muscles and ligaments also occur in response to IVDD and may also contribute to spinal pain, reduced spinal stability and flexibility, and impaired locomotion [32,33,34,35,36]. The ovine spine has been used as a model system for the measurement of neuromuscular responses following spinal manipulation of the normal spine, in spines that had undergone spinal surgery, and in spinal segments containing degenerate IVDs [32,33,34,36,37,38,39,40] (Figure 1).

### 1.2. Importance of Biomechanical Function in IVD Health

The biomechanical function of intervertebral discs is vital for overall spinal health, influencing load distribution, flexibility, nutrient exchange, and degeneration prevention. Mechanical loading on the IVD is known to affect IVD cell metabolism, and excessive loading can lead to degenerative changes [41]. The biomechanical properties of the intervertebral disc (IVD) rely on the ongoing biosynthetic activity of IVD cells and their regulation of extracellular matrix degradation. With aging or excessive trauma, the degradation of proteoglycans and collagen occurs, leading to a decrease in the fixed charge density and dehydration of the nucleus pulposus (NP). This can alter compressive stiffness and disc height, thereby increasing the chance of prolapse [42]. The gelatinous NP is ultimately replaced by fibrocartilaginous tissue, leading to disc failure as the annulus fibrosus (AF) cannot withstand the complex compressive and shear forces. Pathological conditions such as trauma and age-related degeneration have been identified as significant contributors to chronic lower back pain. Additionally, disruption of the IVD can result in spinal instability, potentially causing deformity or neurological dysfunction [42].

### 1.3. Overview of Orthobiological Strategies

Orthobiologics represent a promising frontier in musculoskeletal medicine, offering innovative solutions for enhancing healing and recovery. By harnessing the body’s natural biological processes, these therapies aim to improve outcomes in a range of orthopedic and spinal conditions. The most used orthobiologics in orthopedic surgery such as bone grafts, cell therapies, osteoinductive growth factors, and proteins are ready to use in clinical settings [43]. However, the indiscriminate use of orthobiologics and the lack of clear guidelines for their processing and delivery could jeopardize patient health and safety. Therefore, more rigorous research is necessary to establish the optimal indications and applications for orthobiologics. Not only should these treatments be safe, but they must also provide tangible benefits to patients. Given their crucial role in tissue structure, function, and regeneration, proteoglycans (PGs) form an important component of orthobiologics. They contribute to the body’s natural healing processes and are increasingly being explored for therapeutic applications in regenerative medicine [44,45,46,47,48].

The aim of this review is to explore potential therapeutic opportunities with engineered biomimetic proteoglycans to specifically target the mechanically incompetent degenerate IVD depleted of its space-filling aggrecan proteoglycan [49].

## 2. Proteoglycans Relevant to IVDs

PGs are major components of the extracellular matrix (ECM) in IVDs, primarily found in the NP, AF, and cartilaginous endplate (CEP). They contribute to the disc’s structural integrity, helping to maintain its shape and mechanical properties. PGs also help distribute mechanical loads evenly across the IVD, enabling it to absorb shock during activities like walking, running, and lifting [50,51]. Along with this, PGs play a crucial role in cell signaling, influencing the proliferation, differentiation, and metabolism of the IVD [52]. Aggrecan is the most abundant proteoglycan in the IVD tissues. It is highly hydrophilic and attracts water, which is essential for maintaining the hydration of the NP [53]. This hydration is crucial for the disc’s ability to absorb shock and resist compression. Multiple aggrecan molecules can form aggregates with hyaluronic acid, further enhancing their ability to bind water and provide mechanical support (Figure 2).

Versican is another widely distributed PG in the NP, the depletion of which could be a hallmark of tissue fibrosis [54]. Decorin, a small class 1 leucine-rich PG, has shown protective effects on IVDs from apoptosis by stimulating autophagy via mTOR signaling [55]. Like decorin, biglycan is another small leucine repeat proteoglycan (SLRP). Studies have reported that deficiency in biglycan can accelerate disc degeneration through progressive reduction in the NP area [56]. Lumican is a class II SLRP that has been suggested as a potential discriminative biomarker for IVDD and LBP [57]. Interfering with lumican significantly reduced the TNF-α-induced inflammatory response, cell cycle arrest, and cellular senescence. It has been elucidated that the ASK1/p38 pathway is involved in lumican-mediated changes in NP cells via FasL [58]. In conditions such as DDDs, the quantity and functionality of PGs decline, leading to reduced hydration and structural integrity and ultimately leading to pain and dysfunction. Some key naturally occurring PGs crucial for the structure and function of the IVD are outlined in Table 1.

While natural PGs are essential for maintaining the health of IVDs, with increasing age, their synthesis in the human body decreases, leading to dehydration and loss of structural integrity of the IVD. PGs are also susceptible to enzymatic degradation by matrix metalloproteinases (MMPs) and aggrecanases, particularly in degenerative conditions. This loss of function contributes to the breakdown of the disc structure. Such limitations in the context of degeneration highlight the need for innovative therapeutic approaches, such as engineered PGs (biomimetic) to restore disc function and alleviate symptoms associated with degeneration.

## 3. Biomimetic Engineering of Proteoglycans

Biomimetic engineering of PGs involves the design and creation of synthetic or modified proteoglycans that replicate the structural and functional properties of natural proteoglycans. This approach aims to harness the beneficial roles of PGs in biological systems while overcoming their limitations. The core protein backbone can be engineered to match or improve upon the properties of natural PGs. This may include modifications to enhance binding with other matrix components or cells. GAG chains can be customized in terms of type (e.g., chondroitin sulfate and hyaluronic acid), length, and branching patterns [65]. These modifications can significantly influence hydration capacity and biological activity. Advanced methods such as click chemistry or enzymatic synthesis can be used to attach GAGs to core proteins, ensuring precise control over the final structure [66].

## 4. Development of Biomimetic PGs to Modulate IVDD Physiology

While ECM scientists have been interested in GAGs and PGs for decades, it is only relatively recently that the immense potential of PGs to modulate tissue environments and regulate resident cell populations and migratory progenitor stem cell populations has been appreciated [67,68,69,70,71,72,73,74,75]. The multifunctional properties of PGs, their ability to bind and sequester growth factors and cytokines and interact with an extensive range of ECM molecules, regulate cellular signaling events and physiological processes, and stabilize weight-bearing and tension-resisting tissues make them extremely attractive effector molecules with therapeutic potential in multiple connective tissue diseases.

IVDD and discogenic LBP accompany internal disruption to normal IVD architecture and loss of major IVD PG aggrecan. With mechanical overload of the degenerate IVD, disruption in CEP structure can also occur [76,77,78,79] and, in turn, be a significant contributor to the generation of LBP [80]. The CEP is densely innervated by the basivertebral branches of the sinuvertebral nerve, and pathological ingrowth of these nerves into the CEP from the vertebral body occurs during IVDD, leading to pain generation. S100, PGP9.5, and substance P have all been immunolocalized in CEP nerves [81,82,83,84,85]. GAP43, PGF9.5, and glial fibrillary acidic protein have also been immunolocalized in ingrowing nerves into the degenerate AF [49] associated with vertebral remodeling adjacent to the annular defect site [86]. Degeneration of an IVD disc can induce neurogenic inflammation in adjacent healthy discs [87] and this can potentiate nociceptor sensitization. Neurogenic inflammation is induced by neuropeptide release from activated primary afferent terminals of multisegmented dorsal root ganglion neurons following injury to the AF [88].

Biomimetic GAG and PG biotherapeutics are of considerable interest in cartilage repair strategies [89,90] and methods have been developed for the production of biomimetic PGs [91]. Various formats have been examined including hydrogels and nanoparticles in the development of multifunctional drug delivery systems [92]. A thorough examination of the nanomechanical properties of aggrecan clearly establishes it as a molecule of central importance in the functional properties of cartilages [67]. The development of biomimetic aggrecan is aimed at reproducing the properties of native aggrecan in cartilaginous tissues. Many forms of GAG bio-scaffolds have been developed and shown to have utility in a diverse range of tissue repair strategies and they have provided invaluable insights as to how specific GAGs direct cellular behavior to affect connective tissue repair processes [68,69]. In an effort to obtain the bottle-brush structure of aggrecan, a co-polymer mimetic aggrecan was prepared using CS chains attached to carboxymethyl cellulose as a PG core protein mimic and cross-linked with modified chitosan, which served as an HA substitute, in order to reproduce an aggrecan–HA ternary complex type structure [93,94]. Glycopolymers have also been prepared whereby an HA-binding peptide (GAHWQFNALTVRGGGC) and a type II collagen peptide (WYRGRL) were attached to CS.

mLUB mimics the proteoglycan lubricin which, in cartilage, acts as a boundary lubricant but, in fibrocartilages, lubricates collagen fibril bundles [95,96,97,98,99]. Periodate oxidation cleaves between vicinal hydroxyl groups in glucuronic acid residues in the CS chainm, generating reactive aldehydes to which these peptides are attached using the bifunctional cross-linking agent BMPH (*N*-[*β*-maleimidopropionic acid] hydrazide) (Figure 3). This produces mAGC, an aggrecan biomimetic co-polymer containing an HA-binding peptide, a further glycopolymer mimicking lubricin containing HA-binding and type II collagen peptides. mAGC CS chains with attached HA-binding peptides mimic the HA-binding properties of native aggrecan [100] and have superior water regain properties to native aggrecan, and mAGC constructs have superior compressive strength (78% increase). The catabolism of mAGC constructs by MMP-13 and ADAMTS-5 [101,102] is also reduced compared to animal models of OA [101,103]. mAGC functionally mimics aggrecan but lacks the known cleavage sites of native aggrecan, protecting the molecule from proteolytic degradation. A peptidoglycan lubricin-mimetic, mLUB15, has also been prepared using a similar methodology to that used for mAGC using an HA-binding peptide (GAHWQFNALTVRGGGC) and a type II collagen-binding peptide (WYRGRL) [95].

In an alternative approach, the preparation of graft co-polymer biomimetic PGs to mimic the bottle-brush-like structure of aggrecan involves the attachment of CS chains to a synthetic poly(acryloyl chloride) backbone [104,105]. Large and small PG mimetics, range from a 10 kDa polyacrylate core with ~7–8 CS chains attached (BPG10) [105] to a 250 kDa core polymer with ~60 CS chains attached [104]. The water-imbibing properties of these biomimetic PGs exceed that of aggrecan and unconjugated CS [105]. In addition to their favorable swelling properties in tissues, these polymers also regulate collagen fibril formation when mixed into a collagen gel, as do SLRPs [106]. Furthermore, the BPGs associate with pericellular native PGs in OA cartilage and aid in ECM repair processes. The polyacryloyl backbone of BPG10 is not prone to enzymatic degradation [107] increasing the biological half-life of biomimetic PGs in a tissue environment (Figure 4). A number of strategies have been used with BPG10 in cartilage repair studies in which it was found to reproduce the macromolecular architecture of normal cartilage and had water uptake properties superior to natural proteoglycans [105]. An additional feature that could be built into biomimetic PGs is the incorporation of a bioactive module for use as a drug delivery vehicle [108,109]. Methods are now available to edit the chemical structure of such PGs, and bioactive modules can now be incorporated to produce novel biotherapeutic polymers [110]. Furthermore, methods are also available for controlled glycan sequences of GAG chains on engineered proteoglycans [111,112,113,114,115,116,117,118].

PG and GAG hydrogels have been used as drug carriers [108] and for the delivery of growth factors and cytokines [72,119]. Table 2 shows how specific growth factors delivered via biomimetic PG systems enhance IVD regeneration by stimulating cell proliferation and matrix production and reducing inflammation. Aggrecan mimetic nanoparticles have also been used for growth factor delivery [120] and in situ gelling hydrogels for the delivery of BDNF in the repair of spinal cord defects [121]. Furthermore, GAG mimetic polymers can direct mesenchymal stem cell osteo/chondrogenic differentiation to improve cartilaginous tissue repair processes [122]. Mesenchymal stem cells have already been shown to repair annular lesions in a model of experimental IVDD [123], demonstrating the potential of a GAG mimetic–MSC combination therapeutic approach to improve the repair of degenerate IVDs. Biomimetic PGs strengthen the pericellular matrix of normal and osteoarthritic human cartilage [124] helping to re-engineer OA cartilage [125,126]. Biomimetic PGs modify the cartilage pericellular matrix and modulate cell mechanobiology [127] and have similar nano-mechanical properties to native aggrecan [128] (Figure 5).

## 5. Discussion

PG-based biomaterials are designed to replicate the role of native proteoglycans by retaining water, generating osmotic pressure, and providing mechanical support within the disc. Engineered biomimetic PGs often incorporate glycosaminoglycans (GAGs) like chondroitin sulfate, keratan sulfate, or synthetic analogs, to re-establish the matrix’s water-absorbing properties. These materials are capable of increasing hydration and disc height, potentially relieving pressure on surrounding nerves and reducing pain. In addition, the dense negative charge of GAGs enables the attraction of cations, which facilitates osmotic swelling pressure and provides mechanical support under compressive loads, a critical function in load-bearing tissues like the IVD.

Degenerate IVDs may not have the cellular or nutritional capacity to fully replace ECM components that have become degraded and non-functional in the degenerate IVD. However, supplementing these functional components using an orthobiological approach [43,140] may be an effective means of resolving this issue and re-attaining the biomechanical competence of the IVD as a weight-bearing structure, and such studies warrant investigation. Moreover, the use of biomimetic proteoglycans bypasses the requirement of having an adequate nutritional supply or the involvement of resident disc cell populations to drive this process whereby the biomechanical competence of the IVD can be re-attained. However, it may provide a stop-gap measure to allow resident cell numbers to recover from degenerative conditions. Biomimetic PGs such as BPG10 or mAGC may thus be effective polymers to recover a functional IVD extracellular matrix and warrant further investigation.

Furthermore, biomimetic PGs may also be used to deliver drugs, growth factors, or therapeutic modules to the IVD. PGs, due to their GAG chains, bind growth factors through electrostatic interactions, particularly with positively charged growth factors such as transforming growth factor-beta (TGF-β), fibroblast growth factor (FGF), and bone morphogenetic proteins like GDF-6. These interactions allow growth factors to be stored within the proteoglycan structure, protecting them from rapid degradation and allowing for sustained release over time. This controlled release supports prolonged biological activity, which is especially useful in the IVD, where nutrient diffusion is limited.

The design of proteoglycan-based biomaterials includes both natural and synthetic approaches. Natural biomaterials derived from sources such as hyaluronic acid or collagen are biocompatible and can integrate well with native tissue. However, they may degrade too quickly in the harsh IVD environment. In contrast, synthetic proteoglycan mimics—often polymer-based—can be tailored for longer durability and controlled degradation. Some biomaterials incorporate crosslinking techniques to enhance mechanical properties and resist enzymatic degradation, which can extend their efficacy in the disc.

Another promising approach is injectable proteoglycan biomaterials, which allow for minimally invasive delivery into the disc. These injectable formulations can be administered directly into the nucleus pulposus, potentially restoring hydration and preventing further matrix degradation. Advances in hydrogel-based delivery systems enable the controlled release of proteoglycan components, allowing sustained therapeutic effects within the disc space.

Biomimetic proteoglycans (PGs) for treating intervertebral disc (IVD) degeneration offer promise but face several key challenges. First, ensuring biocompatibility and preventing immune reactions are critical, as synthetic PGs may provoke inflammatory responses in the human body. Second, they must resist the IVD’s harsh, enzyme-rich environment without degrading too quickly to maintain efficacy over time.

Mechanical compatibility is another hurdle, as biomimetic PGs need to withstand spinal loading and control swelling to avoid excess pressure and discomfort. Delivering PGs effectively into dense IVD tissue and ensuring that they integrate well with native structures adds complexity, as does the need for even distribution without damaging surrounding tissues. Regulatory approval presents a significant barrier due to the rigorous testing required for safety and efficacy. Long-term studies are essential, given that IVD degeneration is a chronic condition. Furthermore, manufacturing costs are high, and scaling up production may be challenging without clear evidence of long-term benefits.

Addressing these issues—compatibility, durability, integration, regulatory challenges, and cost—will be critical to advancing biomimetic PGs toward clinical trials and eventual therapeutic use.

## 6. Conclusions

This review has illustrated the functional credentials of biomimetic PGs as bioactive agents that stabilize cartilaginous tissues. They can also be functionalized with bioactive modules such as growth factors and bioactive peptides, which are subsequently released when the biomimetic PGs are remodeled in tissues, ensuring specificity of action and efficacy.

As further research and development refine the synthesis and application of biomimetic PGs, their potential in treating a wide range of degenerative diseases, including osteoarthritis and other cartilage-related injuries, becomes increasingly evident. Through ongoing advancements in materials science, biomimetic PGs may evolve into a cornerstone in regenerative medicine, offering a more effective and targeted approach to tissue healing.

## Figures and Tables

**Figure 1 biomimetics-09-00722-f001:**
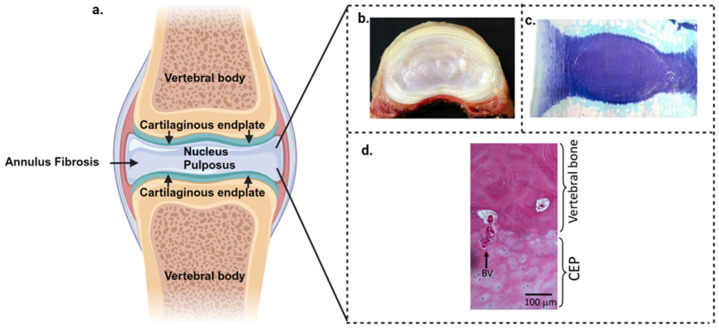
(**a**) Schematic illustrating the major regions of the composite IVD. (**b**) Macro view of the ovine intervertebral disc bisected horizontally showing the radial collagenous layers surrounding a central proteoglycan-rich region known as the nucleus pulposus. (**c**) Vertically sectioned mid-sagittal ovine IVD and adjacent superior and inferior vertebral bodies. Toluidine blue-fast green counterstain. Proteoglycans are stained dark blue. (**d**) Hematoxylin and eosin-stained segment of CEP hyaline cartilage that interfaces with the vertebral bone with chondrocytes distributed throughout the CEP. A blood vessel (BV) penetrating into the CEP is shown with arrows. Created with BioRender.com (accessed on 13 November 2024).

**Figure 2 biomimetics-09-00722-f002:**
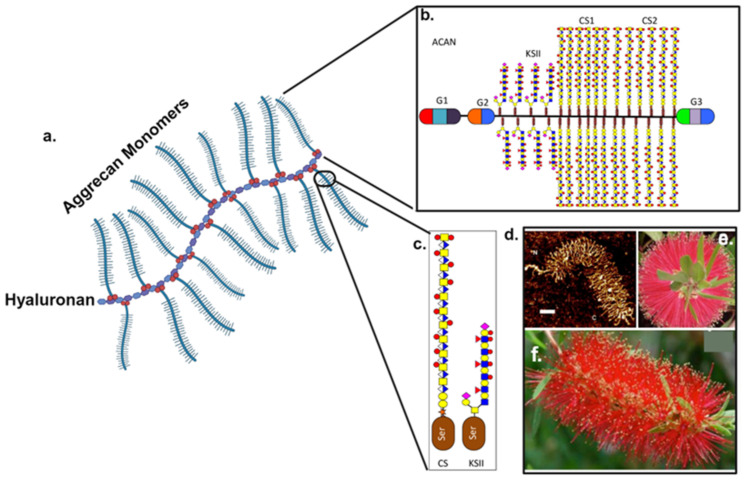
Schematic depiction of the structural organization of native aggrecan, a major proteoglycan of cartilaginous tissues including the IVD showing its 3 globular domains (G1, G2, and G3) and KS- and CS-rich glycosaminoglycan regions (**a**). Details of the CS and KS sidechains of aggrecan (**b**,**c**). Atomic force microscopy (AFM) image of cartilage aggrecan showing its 3D bottle-brush type architecture (**d**), which is well illustrated conceptually by images of the Australian Callistemon bottle-brush shrub (**e**,**f**). The glycan icons used are standard SFNG (Symbol Nomenclature for Glycans) symbols for glycan components. Sulfate groups are depicted as small round red symbols. Created with BioRender.com (accessed on 13 November 2024).

**Figure 3 biomimetics-09-00722-f003:**
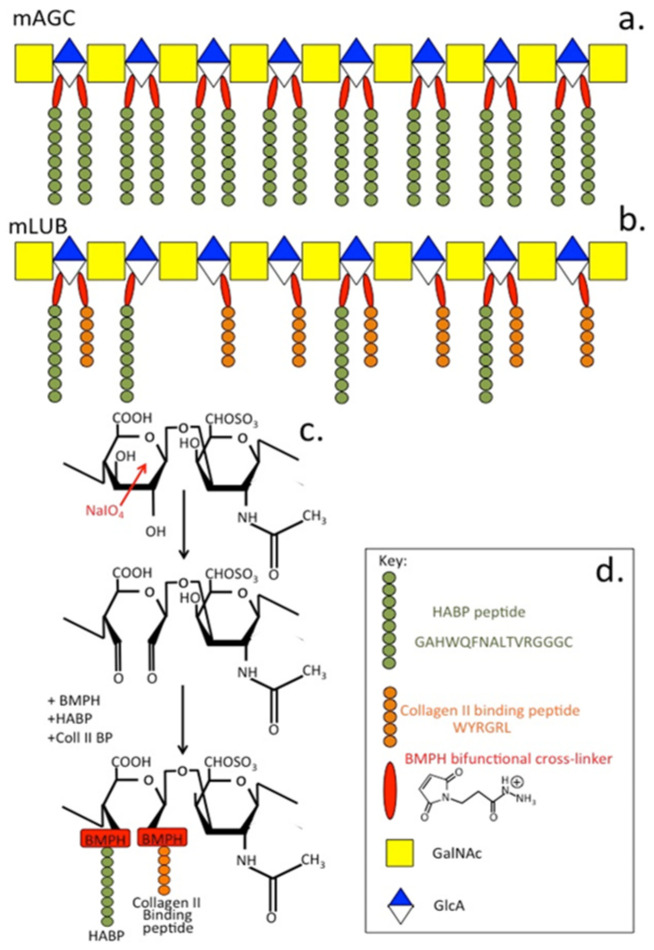
Schematic depiction of the biomimetic proteoglycans mAGC (**a**) and mLUB (**b**) designed to mimic aggrecan and lubricin. (**c**) Assembly of mAGC and mLUB showing cleavage between vicinal hydroxyl groups in the glucuronic acid component of the CS backbone by sodium periodate and attachment of HA and type II collagen binding peptides to the reactive aldehydes so generated using the bifunctional reagent N-β-maleimidopropionic acid hydrazide (BMPH). (**d**) Key for (**c**). Figure modified from [67]. Copyright held by Melrose J.

**Figure 4 biomimetics-09-00722-f004:**
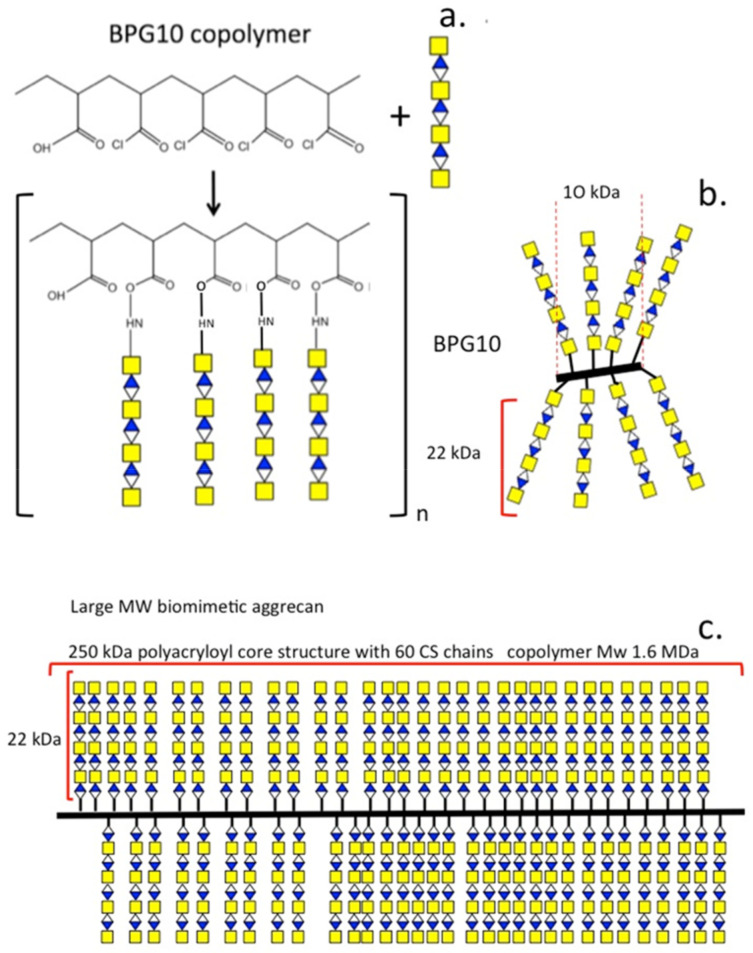
Schematic portrayal of the structure of small and large BPG biomimetic proteoglycans produced by attachment of CS chains to a poly (acryloyl chloride) backbone using bifunctional reagents (**a**) to produce BPG 10 consisting of a 10 kDa core structure with 6–8 attached CS side chains with a similar spacing to that found in native aggrecan (**b**). A large BPG consisting of a 250 kDa core polyacryloyl core structure and 60 attached CS chains was also prepared (**c**). This polymer had a size of 1.6 MDa. Figure modified from [67].

**Figure 5 biomimetics-09-00722-f005:**
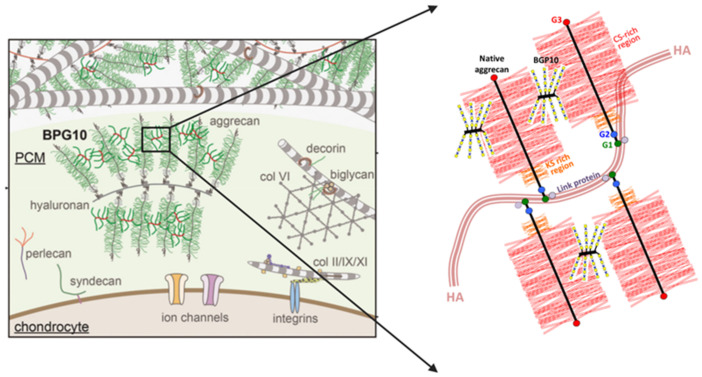
Schematic depiction of native aggrecan–HA aggregates which localize in the chondrocyte pericellular matrix and BPG10 biomimetic proteoglycan that also localizes with them in the pericellular matrix. Figure modified from [127].

**Table 1 biomimetics-09-00722-t001:** Location and functions of PGs in the IVD.

Proteoglycan	Location	Functions	Structure
Aggrecan [53,59]	Nucleus pulposus	- Major component providing hydration and resilience. - Distributes compressive forces. - Maintains disc structure.	- 250 kDa core protein composed of G1, G2, and G3 globular proteins - Chondroitin sulfate (CS) and keratan sulfate (KS) chains between G2 and G3
Versican [54]	Nucleus Pulposus and Annulus Fibrosus	- Involved in cell signaling and ECM organization. - Supports structural integrity. - Plays a role in repair processes.	- 300 kDa core protein - GAG chains of HA and chondroitin sulfate
Decorin [60,61]	Annulus Fibrosus	- Regulates collagen fibril formation. - Influences the mechanical properties of the matrix. - Binds growth factors, modulating their activity. - Regulates cell proliferation and differentiation.	- 40 kDa core protein - Single GAG chain, usually dermatan sulfate - The presence of leucine-rich repeats (LRR)
Biglycan [56,62]	Annulus Fibrosus	- Aids in collagen organization. - Contributes to tissue repair. - Influences matrix properties.	- 42 kDa core protein - LRRs with one or two dermatan sulfate (DS) or chondroitin sulfate (CS) chains
Lumican [63,64]	Nucleus Pulposus and Annulus Fibrosus	- Supports collagen fibril organization. - Enhances the tensile strength of the disc. - Contributes to structural integrity.	- 30 kDa core protein - LRRs with one or two GAG chains, usually dermatan sulfate or keratan sulfate

**Table 2 biomimetics-09-00722-t002:** Biomimetic PGs used for the delivery of growth factors in the intervertebral disc.

Growth Factor	Biomimetic PGs	Effects Observed with Biomimetic PG Delivery	References
Transforming Growth Factor-Beta (TGF-β)	Hyaluronic acid (HA)-based PG mimic	Increased collagen and aggrecan synthesis and improved matrix integrity	[129,130]
Fibroblast Growth Factor-2 (FGF-2)	Chondroitin sulfate (CS)-based PG mimic	Enhanced cell viability and improved matrix production	[131,132]
Bone Morphogenetic Proteins	CS-HA hybrid PG mimic	Restored proteoglycan levels and stimulated matrix repair	[72,133,134]
Insulin-like Growth Factor-1 (IGF-1)	Synthetic GAG analog with controlled release	Enhanced nucleus pulposus cell survival and proteoglycan synthesis	[135,136]
Platelet-Derived Growth Factor (PDGF)	Heparan sulfate (HS)-mimetic PG	Increased cell proliferation and reduced inflammatory markers	[137,138,139]

## Data Availability

No new data were created or analyzed in this study. Data sharing is not applicable to this article.

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
