# Peer review of "Biomimetic Proteoglycans for Intervertebral Disc (IVD) Regeneration"

_biomimetics, 2024, doi:10.3390/biomimetics9120722_

Round 1
Reviewer 1 Report
Comments and Suggestions for Authors
This review paper explores prospective orthobiological strategies using engineered biomimetic proteoglycans to treat degenerate intervertebral discs. Although the topic is appealing, the current version of the manuscript is distant from the requirements for publication in Biomimetics due to several issues:
1) The authors dwell on introducing the content, illustrating biomimetic proteoglycans in one paragraph. Instead, the authors should deepen the discussion about proteoglycan-based biomaterials by dividing the discussion into separate paragraphs. More figures and tables would be helpful for this aim.
2) Table 1: The authors should describe proteoglycans' structure uniformly. Which are the GAGs in the case of aggrecan and biglycan?
3) The authors should highlight the difference between IVD and IVDD since both are defined as intervertebral disc degeneration on page 1, lines 17 and 42.
4) The authors should check and correct typos throughout the manuscript.
Author Response
Comments 1: The authors dwell on introducing the content, illustrating biomimetic proteoglycans in one paragraph. Instead, the authors should deepen the discussion about proteoglycan-based biomaterials by dividing the discussion into separate paragraphs. More figures and tables would be helpful for this aim.
Response 1: Agree. To this point, we have included Table 2 (line 271 and line 289) that shows how specific growth factors have been delivered via biomimetic PG systems to enhance IVD regeneration. Also, we have now renamed point no. 5 (line 286) as “discussion” and have provided more information on proteoglycan-based biomaterials that have been used in IVD
Comments 2: Table 1: The authors should describe proteoglycans' structure uniformly. Which are the GAGs in the case of aggrecan and biglycan?
Response 2: Thanks for pointing this out. We have now included details about the GAGs for aggrecan and biglycan in Table 1(highlighted in red).
Comments 3: The authors should highlight the difference between IVD and IVDD since both are defined as intervertebral disc degeneration on page 1, lines 17 and 42.
Response 3: Agree. Thank you for pointing this out. IVD is for Intervertebral disc and IVDD is for Intervertebral disc degeneration. To avoid any confusion, we have decided not to abbreviate Intervertebral disc (IVD). Therefore, we gave deleted “IVD” from line one.
Comments 4: The authors should check and correct typos throughout the manuscript.
Response 4: The authors have done the same as per the suggestions.
Reviewer 2 Report
Comments and Suggestions for Authors
The review is well-written and does an excellent job of framing the significant socioeconomic impacts of lower back pain. The review should be accepted for publication once my comments below are addressed:
Overall comments and suggestions:
A large part of the review is spent introducing the issues of lower back pain and intravertebral disc degeneration, with the latter half focusing on the intended topic. Overall, the review would benefit from including more depth in describing biomimetic proteoglycans, as I am aware of multiple studies that have not been covered here. In addition, the review cites many other reviews, which detracts from the usefulness of this article. Further, the review frames biomimetic proteoglycans as having great promise for treating intervertebral disc degeneration but offers little explanation as to the hurdles that need to be cleared to see their use in a clinical setting. The concluding statement would benefit from further comments on this matter rather than a single sentence stating further clinical studies are required.
Specific comments:
Title: The title could be more succinct.
Line 80: Should "ability" be "inability" here?
Figure 1: Where did images B, C and D come from? Are these unpublished images provided by the authors?
Line 106: "Collagen type" - did the authors intend to name a specific collagen here? If not, "type" can be deleted.
Line 112-113: This cost is introduced in the introduction so I do not believe it needs to be repeated here.
Lines 127-128: Proteoglycans are being extensively explored in therapeutic applications, the review would be strengthened by including some key references to some of these applications.
Line 189: All the citations included at the end of this statement are other review articles. Including some primary references would strengthen the review.
Line 303-308: Are there any potential drawbacks to using biomimetic PGs to treat IVD? What hurdles remain that prevent their translation into the clinic, or use in clinical trials?
Author Response
Comments 1: Title: The title could be more succinct.
Response 1: Agree. We have changed the title to “Biomimetic Proteoglycans for Intervertebral Disc (IVD) Regeneration “
Comments 2: Line 80: Should "ability" be "inability" here?
Response 2: Agree. Thanks for pointing this out. Modified accordingly in line 80.
Comments 3: Line 106: "Collagen type" - did the authors intend to name a specific collagen here? If not, "type" can be deleted.
Response 3: The intent was not to mention any specific collagen type. Therefore, we have deleted “type “in line 106.
Comments 4: Line 112-113: This cost is introduced in the introduction so I do not believe it needs to be repeated here.
Response 4: Agree. We have deleted the cost related mention in line 112-113.
Comments 5: Lines 127-128: Proteoglycans are being extensively explored in therapeutic applications, the review would be strengthened by including some key references to some of these applications.
Response 5: Agree. This is a great point. To this, we have included key references that highlight their potential as therapeutic agents in regenerative medicine in areas of research relating to Alzheimer’s, ischemic wounds, diabetes and cancer in line 128.
Comments 6: Line 189: All the citations included at the end of this statement are other review articles. Including some primary references would strengthen the review.
Response 6: Agree. We have included original articles from the areas of Cancer and diabetes.
Comments 7: Line 303-308: Are there any potential drawbacks to using biomimetic PGs to treat IVD? What hurdles remain that prevent their translation into the clinic, or use in clinical trials?
Response 7: We have added paragraphs to emphasize the drawbacks of using biomimetic PGs to treat IVD. The changes are reflected in line 338-352. We have also added a few lines (362-367) to the conclusion for an holistic overview of the paper.
Round 2
Reviewer 1 Report
Comments and Suggestions for Authors
The authors have revised and improved the manuscript, changed the title, and tried to deepen the topic.